# Optimized Whole-Mount Fluorescence Staining Protocol for Pulmonary Toxicity Evaluation Using Mouse Respiratory Epithelia

**DOI:** 10.3390/mps8060146

**Published:** 2025-12-03

**Authors:** Richard Francis

**Affiliations:** Cilia Research Laboratory, College of Medicine and Dentistry, James Cook University, Townsville, QLD 4814, Australia; richard.francis@jcu.edu.au; Tel.: +61-7-4781-6065

**Keywords:** ciliated airway epithelia, fluorescence microscopy, immunohistochemistry

## Abstract

A straightforward whole-mount approach has been developed that uses fluorescence imaging, mouse trachea, and a range of off-the-shelf reagents for rapidly evaluating substance toxicity within the ciliated respiratory epithelium. Using this protocol, the lumen of control trachea samples displays a typical cobblestone epithelial structure, a high density of ciliated cells, and minimal evidence of cell death, as visualized by phalloidin, acetylated tubulin, and fixable live/dead staining, respectively. In contrast, trachea subjected to treatments that induce injury show disrupted epithelial architecture and increased cell death, indicating substance toxicity. These results support the utility of this protocol for rapidly detecting and quantifying respiratory epithelial toxicity and differential cell-type susceptibility.

## 1. Introduction

The respiratory epithelium is a thin layer of cells lining the airways and is the primary interface between our lungs and the external environment [1]. This delicate tissue plays a critical defence role by secreting a thin layer of mucus that traps inhaled foreign particles [1,2]. Removal and renewal of this mucus layer is essential for the maintenance of lung health and function [3]; to accomplish this, the respiratory epithelium is covered by motile cilia that beat in an oriented manner to move this mucus up and out of the airway in a process called mucociliary clearance [4]. Due to its fragile structure, the respiratory epithelium is highly susceptible to damage from a wide range of factors [5,6]. Humans are continually exposed to, and often deliberately introduce, novel substances into the respiratory system, such as cigarette smoke [7], vaping aerosols [8], nebulized medicines (both conventional/alternative) [9,10], as well as a wide range of environmental pollutants, dusts, and combustion byproducts [11,12]. The wide range of possible novel inhaled materials poses a challenge for respiratory health research and requires standardized methods to assess potential lung toxicity.

Current approaches for assessing airway cilia cell biology generally fall into two categories: acute ex vivo methods and extended in vitro culture systems. Ex vivo techniques include using freshly harvested human nasal or airway brush biopsies [13,14,15], while in vitro approaches commonly involve growing isolated airway epithelial cells at an air–liquid interface to promote differentiation and reciliation [16,17]. Despite their widespread use, both strategies have notable limitations. Brush biopsy sampling removes ciliated epithelial cells from their native tissue context, which can lead to cellular injury [18]. In contrast, air–liquid interface culture systems are resource-intensive and time-consuming, requiring specialized consumables and prolonged culture periods, often 4–6 weeks, before mature/functional ciliated cells suitable for analysis are obtained [19]. The protocol described here provides a technically straightforward method to quickly assess substance toxicity within the ciliated respiratory epithelium. This whole-mount tracheal preparation preserves epithelial architecture and native signalling environments, making it more physiologically relevant than isolated in vitro cultures. This approach also enables relatively high-throughput analysis of multiple animals or tissue samples at once and is readily adaptable to investigate how various factors, including pharmacological agents, genetic modifications, and environmental exposures, can affect ciliated epithelial cell survival.

Possible limitations of this protocol include the use of ex vivo whole mount tissue preparations, which do not fully replicate the complex physiological environment of the respiratory epithelium in vivo, which may influence epithelial function and responses to injury or treatment. Additionally, the use of animal tissue introduces potential species-specific differences, which may limit direct translation of findings to human physiology. Thus, validation using human respiratory epithelial samples or complementary in vivo models may be required to confirm the relevance of any potential findings.

## 2. Experimental Design

### 2.1. Materials

All tools/reagents purchased for this protocol are listed in Table 1; reagents needed to be made are listed in Table 2.

### 2.2. Equipment

Dissection Microscope with light source. Allowing up to ~8–10× magnification, e.g., Leica EZ4 (Leica, Wetzlar, Germany); Zeiss Stemi 305 (Zeiss, Oberkochen, Germany); Olympus SZX7 (Olympus, Tokyo, Japan); Nikon SMZ800N (Nikon, Tokyo, Japan).Rocking Plate, e.g., BenchRocker variable 2D rocker (Sigma-Aldrich, St. Louis, MO, USA).Fluorescent Microscope [20]: Upright or inverted; high magnification/NA (≥40×/>1 NA) water/glycerol or oil immersion objective; capable of epifluorescence or confocal fluorescence imaging. Required fluorescence filters: BLUE (Ex 405 nm/Em 451 nm), FITC (Ex 492 nm/Em 520 nm), TRITC (Ex 540–545 nm/Em 570–573 nm), e.g., Zeiss LSM 710 (Zeiss, Oberkochen, Germany).

## 3. Procedure

### 3.1. Collection of Mouse Tracheas

Care should be taken that animals used are of the same sex, genetic background, and approximate age to minimize possible uncontrolled variables. An exhaustive illustrated step-by-step outline of the mouse trachea harvest can be found in a previous methods protocol [21], in brief:Euthanize the mouse using CO_2_ euthanasia.Use blunt forceps and dissecting scissors to isolate and remove the mouse pluck (heart/lungs/trachea) from the animal.Place the mouse pluck into a 35 mm culture dish filled with 4°C PBS onto the stage of a dissection microscope.Use micro scissors and tweezers to separate the trachea from the heart, lungs, esophagus, and all fat/miscellaneous tissue from the trachea surface.Use a P1000 pipette and P1000 tip to gently flush the tracheal lumen with PBS to remove any blood/mucus.Use micro-scissors to cut the trachea into short lengths (>4–5 cartilaginous rings in length), then, using two sagittal cuts, divide the trachea lengths into half trachea sections.

### 3.2. Trachea Treatment and Fluorescent Staining

Each reagent change listed below (e.g., wash/staining) requires the previous liquid to be removed from each sample. This may be achieved through simple aspiration using a P1000 pipette, or, if using a Corning NETWELL 12-well plate, each sample containing basket can be lifted and placed into a new well with fresh liquid. Each wash only requires a liquid change (~30 s); longer incubation times are listed. For incubation steps, samples should be placed onto a rocking plate and gently rocked.

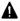
 **CRITICAL STEP:** Care should be taken that trachea samples are not floating on top of the liquid when washing/staining and should be completely submerged.

#### 3.2.1. Sample Treatment

Trachea sections are placed into 12-well plates; multiple sections can be placed per well. Recommended one well per experimental treatment containing ~5–6 trachea sections (from 5 to 6 animals), the first well is reserved for sham/control-treated samples. As 4+ trachea sections can be harvested per mouse trachea, each animal can provide a control sample and 3+ different possible sample treatments (e.g., different chemical concentrations).**OPTIONAL STEP:** Instead of using a standard 12-well plate, 12-well plates with mesh-bottomed inserts can be used (e.g., Corning NETWELL plates). This will allow easier tissue processing by simply moving inserts/tissue between wells for washing/staining.Trachea samples are then exposed to the treatment being assessed for toxicity (e.g., 10 min of incubation with varying concentrations of H_2_O_2_ [9]).

#### 3.2.2. Live/Dead Staining

Live/dead staining is carried out using the ThermoFisher Live/Dead fixable violet dead cell stain kit, as outlined below.

Wash samples three times with PBS (1–2 mL per well/per wash).Dissolve the live/dead dye in DMSO as per kit instructions (50 μL of DMSO added to one vial of dye, mixed well, and visually inspect to confirm all dye has dissolved).Add 1 μL of the reconstituted fluorescent dye to 1 mL of PBS, mix, then add to each sample well (1 mL dye/PBS per well);
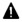
 **CRITICAL STEP:** After the addition of fluorescent dye, samples should be protected from light for all subsequent steps. This can be performed by wrapping the 12-well plate in aluminum foil during incubations.Incubate samples with fluorescent dye at room temperature for 30 min.Wash samples once in PBS (1–2 mL per well).Fix samples in 4% PFA at room temperature for 15 min.Wash samples three times in PBS (1–2 mL per well/per wash).
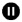
 **PAUSE STEP:** Fixed live/dead stained trachea sections can be stored at 4°C in PBS for 1–2 days before subsequent immunohistochemical staining.

#### 3.2.3. Immunohistochemical Staining

If previously stored at 4°C, allow samples to warm to room temperature.Wash samples twice in PBSFS (1–2 mL per well/per wash).Incubate samples in PBST (1–2 mL per well/per wash) for 10 min at room temperature.Block samples in PBSGS (1–2 mL per well/per wash) for 1 h at room temperature.Wash samples twice in ADB (1–2 mL per well/per wash).Incubate samples with a mouse anti-acetylated tubulin antibody diluted in ADB (1:750 dilution), 1 mL ADB/antibody per well, for 2 h at room temperature.Wash samples three times with PBSGS (1–2 mL per well/per wash).Incubate samples with a FITC-conjugated goat anti-mouse secondary antibody in ADB (1:200 dilution) containing TRITC labelled phalloidin (1 μL/mL), 1 mL of ADB/antibody/phalloidin per well, for 1 h at room temperature.Wash samples once in PBSGS (1–2 mL per well/per wash).Wash samples three times in PBS (1–2 mL per well/per wash).

#### 3.2.4. Sample Mounting for Imaging

The sample mounting preparation is shown in Figure 1. As whole-mount samples are prone to flattening and damage during mounting, a silicon gasket is used to help protect tissue from damage.

Cover a rectangular glass coverslip (24 mm × 50 mm) with ~0.127 mm thick silicone sheet; no adhesive is required, press adhesion is sufficient.


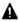
 **CRITICAL STEP:** If the silicon sheet is powder-coated, the silicone should be thoroughly cleaned using cold water, then dried with paper towels before being used to cover a glass coverslip.

2.Create a silicon gasket by cutting and removing a small rectangular section of silicone from the middle of the silicone covering the coverslip using a sharp scalpel, also, trim excess silicone from the edges of the coverslip as needed.3.Place a glass slide (75 × 25 mm) onto the stage of a dissection microscope.4.Using tweezers, place the trachea samples in the middle of the glass slide. Multiple samples can be placed onto the same slide (~5–6 samples per slide).


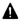
 **CRITICAL STEP:** The trachea samples need to be placed lumen side up on the glass slide. This can be determined by observing trachea curvature under the dissection microscope.

5.Use the edge of a Kimwipe to carefully remove excess liquid from the trachea samples on the glass slide.6.Add a drop of mountant onto the top of each trachea sample (e.g., SlowFade Diamond Antifade Mountant).


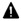
 **CRITICAL STEP:** Care should be taken that no air bubbles are present in the added mountant before the coverslip is placed.

7.Gently lower the silicone-covered coverslip onto the glass slide and trachea samples/mountant, then gently press into place. Care should be taken so that the silicone layer is the middle layer between the coverslip and the glass slide.8.Use a Kimwipe or paper towel to collect any leaked mountant from the sides of the coverslip.9.Seal the coverslip edges using quick-dry nail polish, then air dry at room temperature, protected from light, for 10–20 min.10.
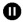
 **PAUSE STEP:** Slides can now be stored at 4 °C before imaging (~1–4 days).

### 3.3. Fluorescent Imaging

Slides are imaged using epifluorescence [22] or confocal fluorescence [23] imaging, depending on the microscopy equipment available. Three fluorescence colour channels are required [24]: BLUE (Ex 405 nm/Em 451 nm), FITC (Ex 492 nm/Em 520 nm), and TRITC (Ex 540–545 nm/Em 570–573 nm). Imaging software will depend on the microscope; it is recommended that a microscope with software that allows the collection of image z-stacks be used. If using epifluorescence, image deconvolution of z-stacks is also recommended.


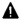
 **CRITICAL STEP:** To minimize fluorescence photobleaching when imaging, do not illuminate samples with bright white or fluorescence excitation light unless actively viewing or collecting images.

Place the glass slide onto the stage of the fluorescence microscope.Use the microscope stage to position the sample in front of the microscope objective and focus using brightfield.


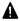
 **CRITICAL STEP:** The coverslip should be the side facing the microscope objective, i.e., if using an upright microscope, the coverslip should be face up. If using an inverted microscope, the coverslip should be face down.

3.Use green (FITC) epifluorescence, microscope eyepieces, stage controller, and focus knobs to quickly scan the sample and find a representative area of ciliated trachea epithelium to image.4.Use the image collection software to optimize fluorescence collection settings (e.g., camera gain/laser power) as needed for each fluorescence channel (BLUE/FITC/TRITC). Image settings should result in images that are bright with minimal overexposure [25].


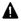
 **CRITICAL STEP:** To optimize image collection, camera gain should be optimized for each different sample area imaged.

5.Collect a three-colour (BLUE/FITC/TRITC) image z-stack. Use the green (FITC) channel to set the top of the image z-stack to just above the top of the cilia. Use the red (TRITC) channel to set the bottom of the image z-stack to just below the epithelial cell layer. The total z-stack depth should be ~12 μm if the epithelial layer is flat. The total number of images to collect within the z-stack will depend on the objective NA [26]. Two example three-colour image z-stacks can be found in the Appendix A.6.Save image z-stacks on computer, making sure to include in file name sample/treatment information.


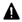
 **CRITICAL STEP:** If saved image files do not contain imaging metadata, make sure to record in the lab book the microscope and camera settings used.

### 3.4. Image Analysis

Fluorescence image z-stacks are analyzed using the FIJI distribution of ImageJ (FIJI 2.3.0/1.53q) [27]. The FIJI distribution of ImageJ is recommended [27] as it contains the Bio-Formats plugin, allowing it to natively open most proprietary image formats used by major microscope manufacturers (e.g., Leica, Zeiss, Olympus, Nikon).

#### 3.4.1. Image Optimization

Open image z-stack using FIJI (FIJI 2.3.0/1.53q).Use the ‘Channels Tool’ (found in /Image/Color/Channels Tool; Shortcut Ctrl+Shift+Z) to set the colour for each channel (Blue for live/dead, Green for FITC, Red for TRITC).Use ‘Z-Project’ to collapse the z-stack into a single image for each colour channel.Use ‘Brightness/Contrast’ to optimize brightness for each colour channel, avoiding overexposure of pixel intensities.Use the ‘Channels Tool’ to display a merged image of all three colour channels.Save the image in TIFF format before cell counting.


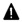
 **CRITICAL STEP:** To open this merged image in non-ImageJ software, the image file should be converted into a single RGB image by using ‘Stack to RGB’ in ImageJ before saving as a TIFF image.

#### 3.4.2. Image Analysis

Open the three colour/channel z-projected image using FIJI (FIJI 2.3.0/1.53q).Use the ‘Multi-point’ tool to manually count the following by clicking on each cell:
Total number of epithelial cells (using the red channel);Total number of ciliated epithelial cells (using the red and green channels);Total epithelial cells stained dead (using the red and blue channels);Total nonciliated epithelial cells stained dead (using the green and blue channels).**OPTIONAL STEP:** A range of automated cell counting plugins exists for ImageJ, which can automate the counting of dead stained cells, e.g., the ‘Cell Counter’ tool (found in /Plugins/Analyze/Cell Counter) or other custom plugins [28,29]. Unfortunately, these tools have not proved reliable in counting total epithelial cells using phalloidin staining, which only labels cell edges, or total ciliated cells using anti-acetylated tubulin antibody staining, which only labels the cilia. Recent advances in AI technology may make automated counting of these cells more practical.**OPTIONAL STEP:** Points can be saved using ‘ROI manager’ as a .roi file.Calculate the following from the above cell counts:
Total number of nonciliated epithelial cells (by subtracting the number of ciliated epithelial cells from the total number of epithelial cells);Total ciliated epithelial cells stained dead (by subtracting the number of nonciliated epithelial cells stained dead from the total number of nonciliated epithelial cells stained dead).
**OPTIONAL STEP:** To control for variability between samples, cell counts can be converted into percentages as outlined below:
Percentage of epithelial cells with cilia (per sample);Percentage of epithelial cells without cilia (per sample);Percentage of total epithelial cells stained dead (per sample);Percentage of nonciliated epithelial cells stained dead (per sample);Percentage of ciliated epithelial cells stained dead (per sample);Percentage of dead epithelial cells stained dead that are ciliated (per sample);
Statistical analysis of group numbers is then assessed using statistics of choice (e.g., ANOVA).

## 4. Expected Results

Representative results obtained from two trachea samples processed using this protocol, including both a control and H_2_O_2_-treated mouse trachea, are presented in Figure 2 and Figure 3 (raw image z-stacks can be found in the online Appendix A). The lumen of control airways should display a characteristic epithelial cobblestone appearance via phalloidin staining (Figure 2A TRITC), containing many ciliated epithelial cells via acetylated antibody staining (Figure 2A FITC). Control airways should display no, or very minimal, dead stained cells using the fixable dead stain (Figure 2A BLUE). Conversely, trachea subjected to toxic injury display perturbations in cell staining (Figure 2B). The magnitude of observed change is dependent on the degree of toxic injury. The example shown in Figure 2B highlights the changes associated with 10 min of incubation with 1% H_2_O_2_ [9], which resulted in the tracheal epithelial layer displaying a perturbed cobblestone arrangement (Figure 2B TRITC) while still containing many ciliated epithelial cells (Figure 2B FITC) and many dead stained cells (Figure 2B BLUE) indicating significant cell death.

Sample analysis (i.e., cell counting) conducted on the images shown in Figure 2A,B revealed a similar number of epithelial cells between the two samples (Figure 2C), which in the control sample was equally split between ciliated and nonciliated (Figure 2C). Conversely, there were slightly more nonciliated epithelial cells in the H_2_O_2_-treated sample, which also displayed a notable elevation in the number of dead stained epithelial cells. When these cell counts are converted into percentages, it is revealed that ciliated epithelial cells display more sensitivity to H_2_O_2_-induced toxicity than nonciliated epithelial cells (Figure 2D).

While Figure 2 demonstrates individual results from two differentially treated trachea samples, Figure 3 presents expected experimental group results quantified from trachea from multiple mice (n = 5 per treatment group, from [9]). Different mouse tracheas were incubated with varying concentrations of H_2_O_2_ (0–1%), then stained according to this protocol. Trachea samples in control and H_2_O_2_-treated groups all showed similar proportions of ciliated vs. nonciliated epithelial cells (Figure 3A). Live/dead staining revealed that H_2_O_2_ treatment resulted in a dose-dependent increase in cell death, with ciliated epithelial cells displaying a higher H_2_O_2_-induced toxicity than nonciliated epithelial cells (Figure 3B). Group data were then compared using two-way ANOVA and post hoc Šídák’s multiple comparisons test (Prism 10.3.0; GraphPad Software), which revealed that while there was an apparent trend for increased cell death at higher doses of H_2_O_2_, this only became significant for ciliated epithelial cells subjected to 1% H_2_O_2_.

The observation that ciliated epithelial cells are more sensitive to H_2_O_2_-induced damage than their non-ciliated counterparts highlights their heightened vulnerability to oxidant injury. Recent studies have shown that these ciliated cells express high levels of mitochondrial uncoupling proteins, which limit reactive oxygen species (ROS) production and thereby reduce lipid peroxidation, a marker of oxidative damage [30]. This protective mechanism is essential, as ciliary beating is a highly energy-demanding process requiring substantial ATP production from the numerous mitochondria clustered near the apical surface beneath the cilia [31]. Exposure to H_2_O_2_ likely overwhelms these defences, leading to increased cell death in the metabolically active ciliated cells, while the less energetically demanding non-ciliated cells are comparatively spared.

## 5. Conclusions

In conclusion, this protocol offers a simple and effective way for the rapid evaluation of substance-induced toxicity within the ciliated respiratory epithelium, and enables direct visualization of epithelial structure, ciliary morphology, and cellular integrity without the need for extensive sample preparation or laborious tissue sectioning. Because basic tissue architecture is preserved, subtle effects on cilia and epithelial integrity can be easily observed. This method can be easily adapted to test a wide range of chemicals, drugs, or environmental agents to determine their potential to damage the respiratory epithelium and, as such, may serve as a valuable tool for preliminary toxicity screening in respiratory research.

## Figures and Tables

**Figure 1 mps-08-00146-f001:**
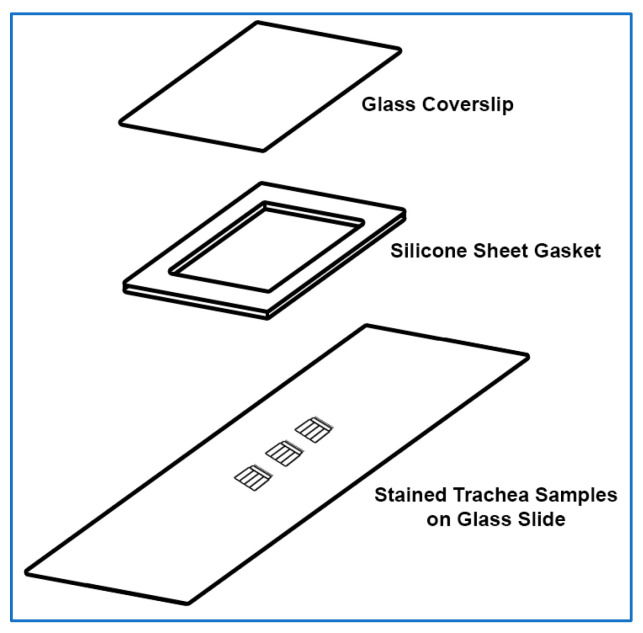
Exploded isometric view of mounted stained trachea samples for imaging. The top glass coverslip (24 mm × 50 mm) and silicone gasket are assembled first. Fixed/stained whole-mount trachea samples are placed in the centre of a glass slide (75 × 25 mm), lumen side up. The edge of a Kimwipe is gently used to remove any remaining PBS from trachea samples on the slide before a small drop of antifade mountant is placed on the top of each sample. The top coverslip is then gently lowered at a 45° angle on the bottom coverslip and gently pressed in place. Finally, the edges of the coverslip are sealed with nail polish.

**Figure 2 mps-08-00146-f002:**
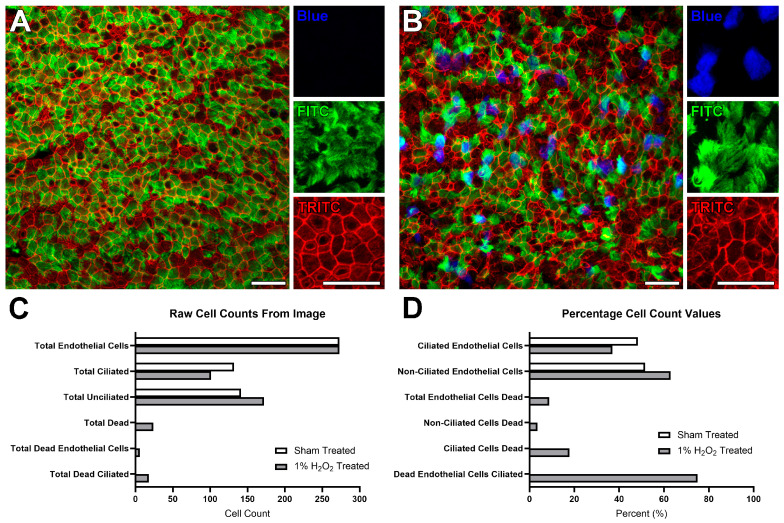
Expected staining results collected using the mice trachea prepared according to the protocol outlined in this study, examples taken from a previously published study [9]. (**A**) A representative fluorescence image generated from a sham-treated trachea. Trachea surface displays characteristic epithelial cobblestone arrangement (Red TRITC channel) containing many ciliated epithelial cells (Green FITC labelled cilia). No labelling was seen in the blue (BLUE live/dead staining) channel. (**B**) A fluorescence image generated from a trachea treated with 1% H_2_O_2_ for 10 min. Trachea epithelial layer displays a perturbed cobblestone arrangement (Red TRITC channel) while still containing many ciliated epithelial cells (Green FITC labelled cilia). Many labelled cells are seen in the blue (BLUE) channel, indicating notable cell death. (**C**) Cell counts from images A and B. Counting reveals a similar number of epithelial cells between the two samples, which in the sham-treated sample was equally split between ciliated and nonciliated. Conversely, there were more nonciliated epithelial cells in the H_2_O_2_-treated sample, which also displayed elevated cell death. (**D**) Numbers counted in the previous panel were converted into percentages. Percentage numbers reveal that ciliated epithelial cells displayed more sensitivity to H_2_O_2_-induced toxicity. Results are presented as raw counts or calculated percentages from each image. Scale bars = 25 μm.

**Figure 3 mps-08-00146-f003:**
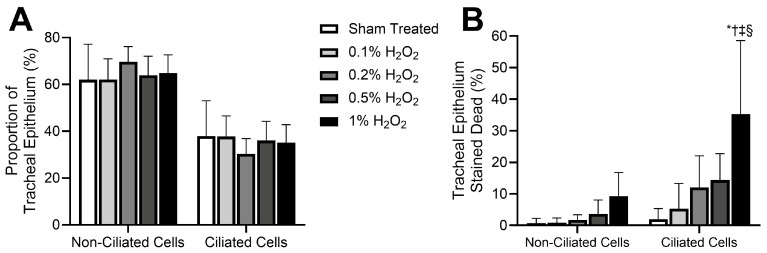
Expected experimental group results quantified from stained mouse trachea prepared according to the protocol outlined in this study (from [9]). Different mouse tracheas were incubated with varying concentrations of H_2_O_2_ (0–1%). (**A**) Trachea samples in the sham- and H_2_O_2_-treated groups all showed similar proportions of ciliated vs. nonciliated epithelial cells. (**B**) Live/dead staining revealed that H_2_O_2_ treatment of trachea resulted in a dose-dependent increase in cell death, with ciliated epithelial cells displaying a higher H_2_O_2_-induced toxicity than nonciliated epithelial cells. Data presented as mean ± SD. * Significantly different from control value (*p* < 0.0001), ^†^ Significantly different from 0.1% H_2_O_2_ dosed group (*p* < 0.001), ^‡^ Significantly different from 0.2% H_2_O_2_ dosed group (*p* < 0.0001), ^§^ Significantly different from time matched 0.5% H_2_O_2_ dosed group (*p* < 0.01).

**Table 1 mps-08-00146-t001:** Tools and reagents needed for this protocol.

Tools	Source	Catalogue#
Cork Dissecting Board	Agar Scientific (Stansted, UK)	AGL4121
#10 Scalpel Blades	Roboz Surgical Instrument Co. (Gaithersburg, MD, USA)	RS-9801
No 3. Scalpel Handle	Roboz Surgical Instrument Co. (Gaithersburg, MD, USA)	65-9843
Blunt Dressing Forceps	Roboz Surgical Instrument Co. (Gaithersburg, MD, USA)	RS-8100
Dissecting Scissors; Straight; 5" Length	Roboz Surgical Instrument Co. (Gaithersburg, MD, USA)	RS-6808
McPherson-Vannas Straight Micro Scissors	Roboz Surgical Instrument Co. (Gaithersburg, MD, USA)	RS-5602
#4 Inox Dumont Tweezers	Roboz Surgical Instrument Co. (Gaithersburg, MD, USA)	RS-4904
Rocking Plate	Thermo Fisher Scientific (Waltham, MA, USA)	88882002
Reagents (to purchase)	Source	Catalogue#
Kimwipes (12 × 21 cm)	Thermo Fisher Scientific (Waltham, MA, USA)	25509-KL
P1000 Micropipette + tips	Thermo Fisher Scientific (Waltham, MA, USA)	4641100N
P100 Micropipette + tips	Thermo Fisher Scientific (Waltham, MA, USA)	4641070N
**OPTIONAL** Repeating Pipette with 25 mL tips	Eppendorf (Hamburg, Germany)	M4 or E3
Phosphate-Buffered Saline (PBS)	Thermo Fisher Scientific (Waltham, MA, USA)	18912014
4% Paraformaldehyde in PBS (PFA)	Thermo Fisher Scientific (Waltham, MA, USA)	J61899.AK
Goat Serum	Thermo Fisher Scientific (Waltham, MA, USA)	16210064
Fetal Bovine Serum	Thermo Fisher Scientific (Waltham, MA, USA)	16000044
Live/Dead fixable violet dead cell stain kit	Thermo Fisher Scientific (Waltham, MA, USA)	L34955
SlowFade Diamond Antifade Mountant	Thermo Fisher Scientific (Waltham, MA, USA)	S36967
Aluminum foil	Thermo Fisher Scientific (Waltham, MA, USA)	040760.HP
35 mm culture dishes	Sigma-Aldrich (St. Louis, MO, USA)	CLS430165
50 mL centrifuge tubes	Sigma-Aldrich (St. Louis, MO, USA)	CLS430829
12-well culture plate	Sigma-Aldrich (St. Louis, MO, USA)	CLS3336
**OPTIONAL** Corning NETWELL 12-well plates	Sigma-Aldrich (St. Louis, MO, USA)	CLS3477
Microscope Glass Slides (75 × 25 mm)	Sigma-Aldrich (St. Louis, MO, USA)	CLS294775X25
Rectangular Glass Coverslips (24 mm × 50 mm)	Sigma-Aldrich (St. Louis, MO, USA)	CLS2975245
Monoclonal Anti-Tubulin, Acetylated antibody	Sigma-Aldrich (St. Louis, MO, USA)	T6793
Phalloidin Peptide-TRITC labelled	Sigma-Aldrich (St. Louis, MO, USA)	P1951
Triton X-100	Sigma-Aldrich (St. Louis, MO, USA)	X100
Fluorescein (FITC) Goat Anti-Mouse IgG	Jackson ImmunoResearch (West Grove, PA, USA)	115-095-003
Revlon Quick Dry Top Coat (nail polish)	Revlon (New York, NY, USA)	-
~0.127 mm thick silicone sheet	AAA Acme Rubber Co. (Tempe, AZ, USA)	CASS-.005X24-65908

**Table 2 mps-08-00146-t002:** Reagents to make fresh on the day of protocol (from items in Table 1).

Phosphate-Buffered Saline with Triton X-100 (PBST): PBS + 0.2% Triton X-100 (100 µL Triton per 50 mL PBS)
Phosphate-Buffered Saline with Fetal Bovine Serum (PBSFS): PBS + 1% FBS (0.5 mL FBS per 50 mL PBS)
Phosphate-Buffered Saline with Goat Serum (PBSGS): PBS + 5% Goat Serum (2.5 mL goat serum per 50 mL PBS)
Antibody dilution Buffer (ADB): PBS + 5% Goat Serum, + 0.1% Triton X-100 (2.5 mL Goat Serum and 50 µL Triton X-100 per 50 mL PBS)

## Data Availability

No new data were created or analyzed in this study. Data sharing does not apply to this article.

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
