# Peer review of "Optimized Whole-Mount Fluorescence Staining Protocol for Pulmonary Toxicity Evaluation Using Mouse Respiratory Epithelia"

_mps, 2025, doi:10.3390/mps8060146_

Round 1
Reviewer 1 Report
Comments and Suggestions for Authors
The protocol is well-written and easy to follow. The tools and reagents needed for the protocol are described in great detail. Some comments to improve the manuscript:
- The drawbacks of the existing protocols are not mentioned in the introduction. Kindly add that.
- In many places, there is no space between units and the number. For example, “35mm culture dishes”. Kindly correct that.
- Line number 126: It is µl/ml and not ul/ml
- × symbol is not being used. Instead x is used. For example-line number 140
- Line number 103: It is written as µL. Please correct to µl.
- Providing the dimensions for Fig. 1 will be useful
- Line number 282: Figure references need to be formatted. Fig 2C
- Line number 211: Kindly indicate where the channels tool is located. Similarly indicate where Z project is located. Please also indicate where the multi-point tool is located.
- It will be useful to discuss why ciliated endothelial cells display more sensitivity to H2O2
- It will also be useful to discuss some automated methods to count cells in Sec. 3.4.2
Author Response
Thanks to Reviewer 1 in providing this useful feedback and allowing me to improve manuscript quality
Comment 1: The drawbacks of the existing protocols are not mentioned in the introduction. Kindly add that.
Response 1: Introduction updated to include potential limitations.
Comment 2: In many places, there is no space between units and the number. For example, “35mm culture dishes”. Kindly correct that.
Response 2: Text updated to add missed spacing between numbers and units
Comment 3: Line number 126: It is µl/ml and not ul/ml
Response 3: Text corrected to fix typo
Comment 4: × symbol is not being used. Instead x is used. For example-line number 140
Response 4: Text corrected to fix typos
Comment 5: Line number 103: It is written as µL. Please correct to µl.
Response 5: Text corrected to fix typo
Comment 6: Providing the dimensions for Fig. 1 will be useful
Response 6: Figure 1 legend text updated to include dimensions
Comment 7: Line number 282: Figure references need to be formatted. Fig 2C
Response 7: Text corrected to fix typos
Comment 8: Line number 211: Kindly indicate where the channels tool is located. Similarly indicate where Z project is located. Please also indicate where the multi-point tool is located.
Response 8: Text updated to include tool location (found in /Image/Color/Channels Tool; Shortcut Ctrl+Shift+Z)
Comment 9: It will be useful to discuss why ciliated endothelial cells display more sensitivity to H2O2
Response 9: Text updated to include discussion on this point
Comment 10: It will also be useful to discuss some automated methods to count cells in Sec. 3.4.2
Response 10: Text updated to include some information on automated cell counting plugins for ImageJ
Reviewer 2 Report
Comments and Suggestions for Authors
This paper describes an advanced, high-resolution research method to obtain precise data on how toxic substances damage the pulmonary structure in an animal model. The introduction is very effective in justifying both the necessity of the protocol (due to environmental hazards) and its technical validity (because it is simple, fast, and utilizes a high-resolution technique on a biologically relevant tissue). The manuscript is well-executed and scientifically sound, and it makes a valuable contribution to the field. I recommend acceptance after minor revisions have been addressed. In particular, regarding the results, I have a concern: while the use of the SEM is acceptable for comparing the means between groups, the SD would have been more informative for describing the actual biological variability of the individual toxic responses among the mice. I suggest the authors explain the rationale for this methodological choice (the use of the SEM) in the Methods section or include the SD alongside the SEM, at least in the caption of Figure 3, to provide a complete picture of the data variability.
Author Response
Thanks to Reviewer 2 in providing this useful feedback and allowing me to improve manuscript quality
Comment 1: while the use of the SEM is acceptable for comparing the means between groups, the SD would have been more informative for describing the actual biological variability of the individual toxic responses among the mice. I suggest the authors explain the rationale for this methodological choice (the use of the SEM) in the Methods section or include the SD alongside the SEM, at least in the caption of Figure 3, to provide a complete picture of the data variability.
Response 1: Figure 3 changed to Mean ± SD as recommended
Reviewer 3 Report
Comments and Suggestions for Authors
The manuscript provides a practical and straightforward whole-mount fluorescence staining method for examining toxic effects on the mouse tracheal epithelium. The protocol is clearly organized, with useful details on reagents, steps, and expected observations. It is likely to be valuable for groups working in airway toxicology. That said, several points require clearer explanation or additional detail to ensure that the method can be reproduced reliably and that its advantages are fully understood.
Comments
-
The authors should expand their discussion of why this whole-mount approach is preferable to other commonly used techniques. It would be helpful to outline the limitations of alternative methods and to explain more clearly what benefits this approach offers, such as maintaining tissue structure, minimizing processing time, or allowing three-dimensional visualization.
-
What specific methodological or conceptual gaps in current airway toxicity assays does this protocol address?
-
Whole-mount preparations can pose challenges when tissues are thick or when imaging depth is limited. Any potential issues caused by mounting pressure or tissue thickness should be acknowledged.
-
Although the manuscript notes that multiple tracheal pieces are taken from each animal, it is not stated how many biological replicates are needed for robust quantitative analysis.
-
The protocol does not comment on expected variability between animals, or whether factors such as age, sex, or mouse strain could influence staining quality or ciliation density.
-
The description of the live/dead staining could be clearer. DAPI is typically used as a nuclear dye, so its role as a dead-cell indicator should be explained more explicitly, particularly if it is part of a specific fixable viability kit.
-
More information is needed on background reduction, quenching, or whether over-staining might produce misleading signals.
-
Whole-mount samples are prone to flattening during mounting. The authors should consider whether pressure from the coverslip might distort ciliary structure or influence the identification of damaged or dead cells.
-
Have automated quantification tools been evaluated? Manual counting can be subjective, so it would be useful to know whether image-analysis algorithms (e.g., thresholding or machine-learning-based segmentation) were tested.
-
The manuscript refers to “treatment being assessed” and briefly mentions H₂O₂, but does not offer guidance on dose ranges, exposure times, or how to ensure even distribution of the treatment across the tissue.
-
Several typographical issues should be corrected. For example, the word “endothelial” appears in places where “epithelial” seems to be intended.
Author Response
Thanks to Reviewer 3 in providing this useful feedback and allowing me to improve manuscript quality
Comment 1: The authors should expand their discussion of why this whole-mount approach is preferable to other commonly used techniques. It would be helpful to outline the limitations of alternative methods and to explain more clearly what benefits this approach offers, such as maintaining tissue structure, minimizing processing time, or allowing three-dimensional visualization.
Response 1: Introduction updated to include other techniques used to study ciliated airway cells, including advantages and limitations of this protocol and others
Comment 2: What specific methodological or conceptual gaps in current airway toxicity assays does this protocol address?
Response 2: This protocol provides an exhaustive step by step description of the whole technical process and is everything a user would need to conduct the whole essay from tissue collection, staining steps, imaging overview, and example data analysis. Refinements in this protocol include the combination of the commercial live/dead stain kit with conventical antibody staining, and the preparation of a silicon gasket to prevent sample crushing during mounting and imaging.
Comment 3: Whole-mount preparations can pose challenges when tissues are thick or when imaging depth is limited. Any potential issues caused by mounting pressure or tissue thickness should be acknowledged.
Response 3: Mouse trachea are uniform in depth and anatomy, are relatively flat, and the epithelial layer is always topmost, no potential issues beyond possible crushing of samples during mounting has been encountered. This is discussed in section 3.2.4: "As whole-mount samples are prone to flattening and damage during mounting, a silicon gasket is used to help protect tissue from damage."
Comment 4: Although the manuscript notes that multiple tracheal pieces are taken from each animal, it is not stated how many biological replicates are needed for robust quantitative analysis.
Response 4: Section 3.2.1 lists the recommended sample size for treatment groups: "recommended one well per experimental treatment containing ~5-6 trachea sections (from 5-6 animals)"
Comment 5: The protocol does not comment on expected variability between animals, or whether factors such as age, sex, or mouse strain could influence staining quality or ciliation density.
Response 5: Section 3.1. updated to include following: "Care should be taken that animals used are of approximately the same age, sex, and genetic background to minimize possible uncontrolled variables"
Comment 6: The description of the live/dead staining could be clearer. DAPI is typically used as a nuclear dye, so its role as a dead-cell indicator should be explained more explicitly, particularly if it is part of a specific fixable viability kit.
Response 6: DAPI staining is not used, DAPI was used to describe the colour channel used to image cells (Ex 405 nm/Em 451 nm). All text has been updated to remove DAPI and use BLUE instead
Comment 7: More information is needed on background reduction, quenching, or whether over-staining might produce misleading signals.
Response 7: This protocol uses robust staining fluorochromes, and if followed users should have no issues with background fluorescence or overstaining. The biggest possible issue could be fluorescence bleaching if care is not taken when sample handling, which is highlighted in section 3.2.2: "CRITICAL STEP After addition of fluorescent dye, samples should be protected from light for all subsequent steps. This can be done by wrapping the 12-well plate in aluminium foil during incubations". The possibility exists for background issues depending on fluorescence microscope used to image samples, but the protocol cannot predict equipment used to image samples. One would assume some level of basic fluorescence imaging experience in the user which would be sufficient to address any possible minor issues arising from their own equipment.
Comment 8: Whole-mount samples are prone to flattening during mounting. The authors should consider whether pressure from the coverslip might distort ciliary structure or influence the identification of damaged or dead cells.
Response 8: Text updated in 3.2.4 to discuss this point, and is why a silicon gasket is used to help protect tissue from damage.
Comment 9: Have automated quantification tools been evaluated? Manual counting can be subjective, so it would be useful to know whether image-analysis algorithms (e.g., thresholding or machine-learning-based segmentation) were tested.
Response 9: Text updated to include some information on automated cell counting plugins for ImageJ (Section 3.4.2)
Comment 10: The manuscript refers to “treatment being assessed” and briefly mentions H₂O₂, but does not offer guidance on dose ranges, exposure times, or how to ensure even distribution of the treatment across the tissue.
Response 10: This protocol is presented as a way of testing any treatment or injury, thus the term “treatment being assessed” is used to be intentionally vague. Full details on the H₂O₂ study are provided in the citation and would not add to understanding of this protocol.
Comment 11: Several typographical issues should be corrected. For example, the word “endothelial” appears in places where “epithelial” seems to be intended.
Response 11: Text updated as suggested
Round 2
Reviewer 3 Report
Comments and Suggestions for Authors
All the requested changes were done by the authors. I recommended publication of this manuscript in its form.